# Molecular Adaptations to Repeated Radiation Exposure in Triple-Negative Breast Cancer: Dysregulation of Cell Adhesion, Mitochondrial Function, and Epithelial–Mesenchymal Transition

**DOI:** 10.3390/ijms26199611

**Published:** 2025-10-01

**Authors:** Noah Dickinson, Alyssa Murray, Megan Davis, Kaitlyn Marshall-Bergeron, Jessica Dougherty, Wuroud Al-Khayyat, Ramya Narendrula, Maggie Lavoie, Emma Mageau, Ronan Derbowka, A. Thomas Kovala, Douglas R. Boreham, Natalie Lefort, Christopher Thome, Tze Chun Tai, Sujeenthar Tharmalingam

**Affiliations:** 1Medical Sciences Division, Northern Ontario School of Medicine University, 935 Ramsey Lake Rd., Sudbury, ON P3E 2C6, Canada; ndickinson@laurentian.ca (N.D.); amurr075@uottawa.ca (A.M.); kaitlyn_mbergeron@outlook.com (K.M.-B.); emageau@laurentian.ca (E.M.); derbowkaronan@gmail.com (R.D.);; 2School of Natural Sciences, Laurentian University, Sudbury, ON P3E 2C6, Canada; 3Health Sciences North Research Institute, Sudbury, ON P3E 2H2, Canada; 4Undergraduate Medical Education, Northern Ontario School of Medicine University, 935 Ramsey Lake Rd., Sudbury, ON P3E 2C6, Canada; nlefort@nosm.ca

**Keywords:** radiation resistance, radiation adaptation, triple-negative breast cancer, TNBC, cell adhesion, mitochondrial dysfunction, epithelial–mesenchymal transition, RNA sequencing, transcriptomic profiling

## Abstract

Radiation resistance presents a significant challenge in the treatment of triple-negative breast cancer (TNBC). To investigate the molecular adaptations associated with radiation therapy resistance, MDA-MB-231 cells were subjected to a repeated radiation (RR) regimen totaling 57 Gy over 11 weeks, followed by clonal selection. The resulting radiation-adapted cells (MDA-MB-231^RR^) were analyzed using whole-transcriptome RNA sequencing, revealing substantial dysregulation of pathways related to cell adhesion, mitochondrial function, and epithelial–mesenchymal transition (EMT). These transcriptional changes were corroborated by functional assays. MDA-MB-231^RR^ cells exhibited reduced expression of adhesion receptors (*ITGB1*, *ITGA2*, *ITGA6*) and extracellular matrix proteins (fibronectin, collagen, laminins), accompanied by significantly impaired cell adhesion to fibronectin, collagen, and laminin substrates. Mitochondrial dysfunction was supported by downregulation of oxidative phosphorylation genes (*MTCO1*, *MTND1*) and confirmed by JC-1 dye assays demonstrating a marked reduction in mitochondrial membrane potential. EMT-associated changes included increased mesenchymal markers and loss of epithelial markers (*CTNNB1*, *SNAI2*, *CK19*), consistent with enhanced migratory potential. Taken together, this study delineates key molecular features of radiation adaptation in TNBC, providing a foundation for the development of targeted therapies to overcome treatment resistance.

## 1. Introduction

Triple-negative breast cancer (TNBC) poses significant clinical challenges due to aggressive tumor behavior, limited therapeutic options, and a propensity for developing resistance to radiation therapy, a key treatment modality for this disease [1,2,3]. Despite advancements in cancer therapeutics, radioresistance remains a critical obstacle, often resulting in treatment failure, recurrence, and poor patient outcomes [4,5]. The molecular mechanisms underlying radioresistance in TNBC remain poorly understood, and identifying these mechanisms is essential to improve therapeutic interventions and patient prognosis [6].

Tumor cells acquire resistance through adaptive responses to repeated radiation exposure, a process referred to as radiation adaptation. This phenomenon involves sustained biological reprogramming that enables cancer cells to withstand therapeutic stress over time. Such adaptations to chronic irradiation are thought to underlie many hallmarks of radioresistance, including enhanced DNA damage repair capabilities [7], modulation of apoptosis pathways [8], alterations in cell cycle checkpoints [9], metabolic reprogramming [10], and epithelial–mesenchymal transition (EMT) [11]. Enhanced DNA repair mechanisms allow cancer cells to efficiently repair radiation-induced DNA damage, leading to increased cell survival following radiotherapy [12]. Additionally, alterations in apoptotic signaling pathways, including the upregulation of anti-apoptotic proteins and suppression of pro-apoptotic signals, further contribute to the resistance phenotype [13,14].

EMT is particularly critical in the context of radiation adaptation, as it facilitates the transition of epithelial cancer cells into mesenchymal phenotypes characterized by reduced cell adhesion, increased migratory and invasive capacities, and resistance to apoptosis [11,15]. Molecular hallmarks associated with EMT include downregulation of epithelial markers such as E-cadherin and cytokeratin 19, coupled with upregulation of mesenchymal markers such as Snail, Slug, vimentin, and N-cadherin [16,17]. These molecular shifts significantly enhance tumor cell plasticity, promoting both metastatic dissemination and resistance to conventional therapies, including radiation [5].

Cellular adhesion and extracellular matrix (ECM) interactions also play pivotal roles in the development of radiation adaptation. ECM components such as fibronectin, collagen, and laminin interact with integrin receptors expressed on tumor cell surfaces, initiating intracellular signaling cascades that influence tumor cell survival, proliferation, migration, and therapeutic resistance [18,19]. Dysregulation of integrin expression, particularly integrin β1 and its binding partners α2, α5, and α6 subunits, has been closely associated with reduced adhesion properties, increased anchorage-independent growth, and enhanced metastatic potential, all of which significantly contribute to treatment resistance [20,21,22]. Beyond ligand-receptor binding, the ECM also acts as a mechanotransducer [23]. Its stiffness and viscoelasticity shape how tumor cells respond to their environment. Cells sense these cues through integrin-FAK/SRC complexes and other mechanosensors [24]. These inputs activate pathways such as YAP/TAZ that regulate survival signaling and DNA-damage responses, thereby promoting radiation and chemotherapy resistance [25].

Metabolic adaptations, particularly those involving mitochondrial dysfunction and metabolic reprogramming, are increasingly recognized as critical contributors to radioresistance in breast cancer [26,27]. Cancer cells commonly exhibit the Warburg effect, characterized by reduced oxidative phosphorylation and enhanced glycolytic metabolism even in the presence of oxygen [28]. This metabolic phenotype provides cancer cells with the energy and biosynthetic precursors necessary for survival and proliferation, especially under conditions of therapeutic stress, such as radiation-induced oxidative damage. Alterations in mitochondrial function, including impaired electron transport chain activity, changes in mitochondrial membrane potential, and increased mitochondrial DNA mutations, further bolster cellular resistance to radiation therapy by reducing susceptibility to apoptosis and enhancing survival under stress conditions [26,27,29].

Despite these known associations, comprehensive profiling of the transcriptomic landscape underlying these adaptive resistance mechanisms in TNBC remains limited [5]. This knowledge gap represents a significant barrier to the development of effective therapeutic strategies targeting radiation-resistant breast cancer cells. To investigate long-term radiation adaptation in TNBC, we established MDA-MB-231^RR^ cell lines by subjecting parental MDA-MB-231 cells to a cumulative 57 Gy of fractionated X-ray irradiation over 11 weeks. Clonal cell populations that survived this regimen were isolated and expanded. These MDA-MB-231^RR^ cells represent radiation-adapted derivatives generated through *Repeated Radiation* (RR) exposure to escalating doses of ionizing radiation and were subsequently subjected to whole-transcriptome RNA sequencing. This unbiased analysis revealed significant dysregulation of genes involved in cell adhesion, mitochondrial function, and EMT-related pathways. To validate these transcriptomic findings, we conducted functional assays assessing ECM-specific cell adhesion and mitochondrial membrane potential, which supported the gene expression trends observed. Collectively, our study provides new insights into the molecular adaptations associated with chronic radiation exposure in TNBC cells and identifies putative targets for further investigation in the context of overcoming therapy resistance.

## 2. Results

### 2.1. Whole-Transcriptome Profiling Reveals Key Pathways Altered in Radiation-Adapted MDA-MB-231^RR^ Cells

To investigate the molecular consequences of repeated radiation exposure, we performed whole-transcriptome RNA-sequencing to compare global gene expression profiles between radiation-adapted MDA-MB-231^RR^ cells and their parental MDA-MB-231^Control^ counterparts. This approach aimed to identify differentially expressed genes (DEGs) that may drive phenotypic alterations associated with radiation adaptation. A total of 13,590 mRNA transcripts were detected across both cell lines. Differential expression analysis identified 1572 DEGs that met the following stringent criteria: fold change > 2 or <−2, FDR-adjusted *p*-value < 0.05, and a minimum average transcript abundance ≥ 40 transcripts per million (TPM). Of these, 547 genes were significantly upregulated and 1025 were downregulated in MDA-MB-231^RR^ cells relative to MDA-MB-231^Control^. The volcano plot presented in Figure 1A visually displays these DEGs, with red and blue circles representing upregulated and downregulated genes, respectively, while grey circles denote transcripts that did not meet statistical significance thresholds. Genes at the plot’s outer edges reflect the most highly dysregulated transcripts, and those higher on the y-axis display the greatest statistical significance. To highlight the most prominent changes in expression, the top 20 upregulated and top 20 downregulated genes are shown in Figure 1B and Figure 1C, respectively, ranked by fold change. Notable upregulated transcripts included genes linked to extracellular matrix remodeling (e.g., *MMP1*), inflammation (e.g., *PTGS2*, *CXCL8*), and transcriptional regulation (e.g., *HEYL*, *GADD45B*). Conversely, strongly downregulated genes encompassed components involved in cell adhesion, immune signaling, and mitochondrial function. The full list of DEGs, including gene names, fold changes, FDR-adjusted *p*-values, and transcript abundance, is provided in Appendix A.

To gain further insight into the biological processes impacted by radiation adaptation, we performed gene ontology (GO) enrichment analysis using iPathwayGuide (AdvaitaBio) to identify significantly overrepresented biological processes among the 1572 DEGs distinguishing MDA-MB-231^RR^ cells from MDA-MB-231^Control^ cells. The top 10 significantly enriched GO biological processes are summarized in Table 1, ranked by FDR-adjusted *p*-value. Processes related to biological adhesion (GO:0022610, FDR = 3.81 × 10^−18^) and cell adhesion (GO:0007155, FDR = 6.50 × 10^−18^) were among the most enriched categories, with over 210 differentially expressed genes mapping to each term. Additional highly enriched processes included multicellular organismal process (GO:0032501, FDR = 4.62 × 10^−16^), regulation of multicellular organismal process (GO:0051239), and anatomical structure morphogenesis (GO:0009653), underscoring broad transcriptional shifts in genes governing cell–cell interactions, tissue organization, and structural remodeling. Furthermore, DEGs were significantly enriched in signaling-related processes such as signal transduction (GO:0007165) and cell communication (GO:0007154), suggesting that radiation-adapted cells undergo substantial rewiring of their intercellular signaling networks. Enrichment in response to stimulus (GO:0050896) further indicates that MDA-MB-231^RR^ cells may have adopted a stress-responsive transcriptional state. Altogether, GO enrichment analysis highlights major biological programs altered during radiation adaptation, particularly those governing cell adhesion, signaling, and tissue-level organization.

To gain deeper insight into the molecular programs driving radiation adaptation, we performed pathway enrichment analysis on the differentially expressed genes using the Kyoto Encyclopedia of Genes and Genomes (KEGG) database via iPathwayGuide (AdvaitaBio). This analysis aimed to identify significantly impacted signaling networks in MDA-MB-231^RR^ cells compared to MDA-MB-231^Control^ cells. The top 10 enriched pathways are listed in Table 2, ranked by statistical significance. Pathways involved in cell adhesion and extracellular matrix (ECM) remodeling emerged among the most significantly enriched. These included the cell adhesion molecules pathway (KEGG:04514), ECM–receptor interaction (KEGG:04512), and protein digestion and absorption (KEGG:04974), each of which featured coordinated downregulation of integrin subunits, collagens, and laminin components. These results suggest that the adhesive interface between cells and the ECM is transcriptionally reprogrammed in radiation-adapted cells. Complementing this, enrichment of the cytokine–cytokine receptor interaction pathway (KEGG:04060), the AGE–RAGE signaling axis (KEGG:04933) and complement and coagulation cascades (KEGG:04610) point to a pronounced pro-inflammatory shift in signaling, with transcriptional changes in multiple immune-modulatory and vascular response mediators. In addition to adhesion and inflammatory pathways, the pathway enrichment analysis identified significant dysregulation in mitochondrial associated signaling cascades. Notably, genes implicated in oxidative phosphorylation and apoptotic regulation were embedded within several enriched pathways, including those related to the AGE–RAGE and cancer signaling axes. These transcriptional changes are consistent with mitochondrial dysfunction and suggest that radiation adaptation may involve remodeling of metabolic and energy production pathways. Together, these findings indicate that radiation-adapted MDA-MB-231^RR^ cells undergo coordinated transcriptional changes affecting cell adhesion, ECM structure, inflammatory signaling, and mitochondrial regulation—hallmarks of cellular adaptation to sustained genotoxic stress.

### 2.2. Radiation Adaptation Alters Expression of Cell Adhesion-Related Genes

To further investigate the dysregulation of adhesion-related pathways identified by RNA-sequencing, we performed targeted RT-qPCR to validate and quantify changes in the expression of key cell–ECM adhesion genes in MDA-MB-231^RR^ cells relative to their parental controls. Genes selected for validation included integrin subunits and structural ECM components (Figure 2). RT-qPCR analysis revealed that several integrin subunits were significantly downregulated in MDA-MB-231^RR^ cells. Specifically, *ITGB1* (Integrin Subunit Beta 1) exhibited a fold change of 0.83 ± 0.03 (*p* < 0.05; Figure 2A), while *ITGA2* (Integrin Subunit Alpha 2), *ITGA6* (Integrin Subunit Alpha 6), and *ITGA10* (Integrin Subunit Alpha 10) were reduced by 0.50 ± 0.04, 0.85 ± 0.06, and 0.57 ± 0.07, respectively (Figure 2B–D). These integrins play critical roles in anchoring cells to ECM proteins such as collagen, laminin, and fibronectin [20], and their downregulation suggests a disruption in integrin-mediated adhesion mechanisms in radiation-adapted cells.

In terms of ECM ligand expression, fibronectin 1 (*FN1*) was significantly upregulated in MDA-MB-231^RR^ cells, with a fold change of 7.39 ± 0.6 (Figure 2E), indicating enhanced production of fibronectin protein, a key scaffold in the ECM that often promotes invasive phenotypes. Conversely, genes encoding ECM components associated with basement membrane integrity were downregulated: laminin subunit beta 1 (*LAMB1*; 0.462 ± 0.03), the alpha 2 subunit of type IV collagen (*COL4A2*; 0.410 ± 0.03) and the alpha 2 subunit of type VI collagen (*COL6A2*; 0.515 ± 0.02) (Figure 2F–H). Collectively, these data confirm that radiation-adapted MDA-MB-231^RR^ cells exhibit significant transcriptional reprogramming of adhesion-associated genes, consistent with impaired integrin signaling and ECM remodeling. These changes may contribute to altered cell–matrix interactions and play a mechanistic role in the radiation-adapted phenotype.

### 2.3. Functional Adhesion to ECM Proteins Is Reduced in MDA-MB-231^RR^ Cells

Given the transcriptional alterations in integrins and ECM components observed in MDA-MB-231^RR^ cells, we next assessed whether these changes translated into functional differences in cell–matrix adhesion. A crystal violet–based adhesion assay was performed to quantify the ability of MDA-MB-231^RR^ and MDA-MB-231^Control^ cells to adhere to wells coated with poly-D-lysine (a non-specific attachment control), fibronectin, collagen, or laminin. Representative images of adherent cells are shown in Figure 3A. On poly-D-lysine, both cell lines demonstrated comparable attachment, indicating no intrinsic defect in general adherence. However, MDA-MB-231^RR^ cells displayed visibly reduced adhesion to all three ECM substrates relative to parental controls. Quantitative analysis (Figure 3B) confirmed that adhesion was significantly impaired on fibronectin (0.410-fold, *p* = 0.0325), collagen (0.574-fold, *p* = 0.0321), and laminin (0.211-fold, *p* = 0.0207) in MDA-MB-231^RR^ cells. These findings functionally validate the gene expression data and indicate that radiation-adapted MDA-MB-231^RR^ cells exhibit impaired engagement with ECM proteins, likely due to integrin downregulation and ECM remodeling. Reduced adhesion may reflect or contribute to changes in cellular motility, invasiveness, or survival in the context of radiation adaptation.

### 2.4. EMT-Associated Transcriptional Changes in MDA-MB-231^RR^ Cells

Given the observed transcriptional dysregulation of ECM components in MDA-MB-231^RR^ cells and the well-established association between ECM remodeling and EMT [30,31], we next investigated whether radiation adaptation induces an EMT-like phenotype. EMT is characterized by the downregulation of epithelial markers and upregulation of mesenchymal traits, facilitating enhanced motility, survival, and therapeutic resistance in cancer cells [5]. RT-qPCR analysis was performed to quantify the expression of canonical EMT markers in MDA-MB-231^RR^ and MDA-MB-231^Control^ cells (Figure 4). Expression of *SNAI2*, a mesenchymal transcription factor known to repress epithelial genes and promote EMT [32], was significantly upregulated in radiation-adapted cells (Figure 4A), suggesting engagement of transcriptional programs that favor mesenchymal transition. However, contrary to expectations for a full mesenchymal transition, the expression of *VIM* (vimentin), a hallmark mesenchymal cytoskeletal gene [33], was significantly downregulated (Figure 4B), indicating suppression of some mesenchymal characteristics. Similarly, *CTNNB1* (β-catenin), involved in adherens junction signaling and often upregulated in the mesenchymal state [34], was also significantly reduced in MDA-MB-231^RR^ cells (Figure 4C). Meanwhile, *CK19* (Cytokeratin 19), an epithelial cytoskeletal marker [35], was significantly downregulated in MDA-MB-231^RR^ cells (Figure 4D), indicating partial loss of epithelial identity. In support of this partial EMT-like shift, we also observed upregulation of *FN1* in MDA-MB-231^RR^ cells (Figure 2E), which is both a structural ECM protein and a hallmark mesenchymal gene [36]. Other EMT-related genes analyzed, including *ACTA2*, *AKT2*, *CD44*, *CDH2*, and *SNAI1*, did not show statistically significant changes in expression. Together, these findings suggest that repeated radiation exposure does not induce a complete EMT phenotype but rather promotes a partial EMT shift, characterized by upregulation of select mesenchymal genes (e.g., *SNAI2*, *FN1*), loss of epithelial markers (e.g., *CK19*), and suppression of other mesenchymal effectors (e.g., *VIM*, *CTNNB1*). This hybrid phenotype may contribute to the altered adhesion and survival capacity of radiation-adapted cells.

### 2.5. Radiation-Adapted Cells Exhibit Altered Expression of Mitochondrial Genes and Reduced Mitochondrial Membrane Potential

RNA-sequencing analysis revealed differential expression of several mitochondria-associated genes in MDA-MB-231^RR^ cells relative to MDA-MB-231^Control^, including downregulation of mitochondrial-encoded transcripts such as *MTCO1*, *MTND1*, and *MTCO1P12*. Given the critical role of mitochondria in regulating energy production, oxidative stress responses, and apoptosis—all processes known to influence radiation sensitivity [29,37]—we sought to validate and functionally characterize mitochondrial alterations in the radiation-adapted phenotype. Here, we first examined the expression of key mitochondrial genes by RT-qPCR. As shown in Figure 5 and matching the RNA-sequencing results, expression of the mitochondrial-encoded genes *MTCO1*, *MTCO1P12*, and *MTND1* was significantly downregulated in MDA-MB-231^RR^ cells relative to MDA-MB-231^Control^ (*p* < 0.05). Conversely, two nuclear-encoded genes involved in mitochondrial transcription and translation, *MTERF4* and *MTRF1L*, were significantly upregulated. No significant differences were observed in the expression of *MTCYB* or *MTND6*. These findings suggest that mitochondrial gene expression is dysregulated at both the mitochondrial and nuclear levels in radiation-adapted cells.

To assess whether these transcriptomic changes were associated with functional alterations, we measured mitochondrial membrane potential using the JC-1 dye assay [38]. In healthy, polarized mitochondria, JC-1 accumulates and forms J-aggregates that emit red fluorescence, whereas in depolarized mitochondria, JC-1 remains in its monomeric form, emitting green fluorescence. As shown in Figure 6A, MDA-MB-231^RR^ cells exhibited a marked shift in JC-1 fluorescence, with a reduced red signal (reflecting fewer polarized mitochondria) and increased green signal (indicating elevated mitochondrial depolarization) compared to MDA-MB-231^Control^. Quantification confirmed a significant reduction in J-aggregates (*p* < 0.05), a corresponding increase in J-monomers (*p* < 0.05), and a nearly 50% decrease in the J-aggregate-to-J-monomer ratio (*p* < 0.05), indicative of compromised mitochondrial membrane potential in the radiation-adapted cells (Figure 6B–D). Taken together, these data demonstrate that MDA-MB-231^RR^ cells exhibit transcriptional dysregulation of mitochondrial genes alongside functional mitochondrial impairment, suggesting that mitochondrial adaptation may contribute to the long-term survival and stress tolerance of cells repeatedly exposed to radiation.

## 3. Discussion

Repeated radiation exposure in TNBC cells drives widespread transcriptional and functional reprogramming. Transcriptomic analysis of MDA-MB-231^RR^ cells revealed widespread dysregulation across multiple cellular pathways, including those related to adhesion, mitochondrial function, and mesenchymal transition. These transcriptomic changes were corroborated by targeted validation using RT-qPCR and supported by functional assays. Specifically, radiation-adapted cells displayed reduced expression of key adhesion molecules, consistent with impaired cell–ECM interactions, and exhibited diminished adhesion to fibronectin, collagen I, and laminin substrates. Mitochondrial membrane potential assays revealed significant depolarization, aligning with RNA-seq findings that indicated suppression of oxidative phosphorylation and TCA cycle genes. Additionally, gene expression analysis identified features of partial EMT, including upregulation of *SNAI2* and *FN1* alongside downregulation of *CK19*, while classical mesenchymal markers such as *VIM* and *CTNNB1* were unexpectedly reduced. Together, these results indicate that radiation adaptation is characterized by complex remodeling of adhesion dynamics, energy metabolism, and phenotypic plasticity, revealing interconnected pathways that may support tumor persistence after radiotherapy.

### 3.1. Remodeling of Cell–ECM Engagement and Partial EMT Under Radiation Stress

Across RNA-sequencing, GO, and pathway analyses, adhesion emerged as the most coherent signature of radiation adaptation. Adhesion-related terms dominated GO (Table 1), and KEGG modules such as ECM–receptor interaction and cell adhesion molecules were significantly enriched (Table 2). The aggregate transcriptomic pattern argues for a reprogramming of cell–matrix engagement rather than a uniform gain or loss of adherence: integrin subunits that confer high-affinity binding to laminin and collagen (*ITGA2*, *ITGA6*, *ITGB1*, *ITGA10*) were reduced, while the matrix blueprint shifted toward increased *FN1* expression with concomitant decreases in *LAMB1*, *COL4A2*, and *COL6A2* (Figure 2; Appendix A). Functionally, MDA-MB-231^RR^ cells adhered less to fibronectin, collagen, and laminin (Figure 3), indicating that the transcriptional rewiring is phenotypically expressed at the level of substrate engagement.

These features are consistent with adhesion plasticity in which cells downshift stable integrin–ECM anchorage while remodeling the matrix they deposit [22]. Reduced β1- and α2/α6-family integrins would be expected to blunt focal-adhesion maturation on collagen and laminin, lower traction forces, and attenuate canonical outside-in signaling [39]. In parallel, elevated FN1 suggests a shift toward autocrine fibronectin assembly and provisional matrix remodeling [40]. Such decoupling of receptor repertoire and matrix composition can promote dynamic attachment–detachment cycles, enabling cells to tolerate fluctuating microenvironments and to navigate damaged stroma after irradiation [41,42]. It also provides a plausible route to anoikis tolerance [43]; weakening of stable adhesions can be offset by transient, context-dependent contacts and by matrix self-provisioning, both of which are compatible with survival under radiation stress [44].

The EMT axis intersects with this adhesion reprogramming in a noncanonical way [31]. Expression of *SNAI2* was increased and *CK19* was decreased, pointing to a loosening of epithelial identity, yet expressions of *VIM* and *CTNNB1* were reduced (Figure 4), arguing against a full mesenchymal conversion. This partial-EMT configuration aligns with the adhesion phenotype: epithelial determinants are relaxed enough to permit motility and matrix remodeling, but a full-scale mesenchymal cytoskeleton is not installed. In practical terms, such a hybrid state can support collective or amoeboid-like modes of movement and reduce dependence on long-lived focal adhesions, behaviors that would be advantageous during repeated radiation cycles [45].

Extending this logic, repeated radiation appears to select an invasion-permissive transcriptional program. For example, a protease–inhibitor imbalance is evident, with induction of *MMP1* (matrix metalloproteinase-1) and *MMP3* (matrix metalloproteinase-3) together with suppression of *TIMP2* (tissue inhibitor of metalloproteinases-2; Appendix A); this pattern is consistent with accelerated basement-membrane and interstitial-matrix turnover [46]. In parallel, downregulation of laminin/collagen scaffolds (*LAMB1*, *COL4A2*, *COL6A2*) together with upregulation of *FN1*, *VCAN* (versican), *TNC* (tenascin C), and *SPOCK1* (testican-1) indicates a shift toward a provisional, growth factor-rich matrix that supports motility and survival [47,48]. In addition, increased *HAS2* (hyaluronan synthase 2), acting with upregulated *VCAN*, implies enhanced hyaluronan synthesis and a hydrated pericellular coat that promotes cell scattering [49]. Moreover, upregulation of *DDR2* (discoidin domain receptor 2) and *RHOC* (Ras homolog family member C) aligns with heightened collagen sensing and contractility-driven invasion [50]. At the same time, elevated *MFGE8* (milk fat globule-EGF factor 8) with reduced *ITGA6*/*ITGB4* suggests weakened hemidesmosomes and compensatory reliance on αvβ3/αvβ5 interactions on fibronectin or vitronectin [51,52]. Finally, *CXCL8* (interleukin-8) upregulation is consistent with microenvironmental conditioning that supports vascular access and early metastatic seeding [53]. Collectively, these alterations delineate a pro-invasive, metastasis-competent state that emerges in parallel with adhesion and ECM plasticity in radiation-adapted TNBC cells.

In line with a stress-adaptive adhesion program, *NDRG1* was upregulated in MDA-MB-231^RR^ cells (2.6-fold). Recent work shows that NDRG1 is PKC-responsive under acute stress in breast cancer [54]. Here, NDRG1 promotes breast cancer invasion via ROCK1–cofilin phosphorylation signaling and enhances stress-fiber assembly. NDRG1 also upregulates ECM-reorganization genes. These functions align with the adhesion/plasticity phenotype observed in this study and nominate *NDRG1* as a candidate node for future mechanistic testing.

Overall, the data presented in this study position adhesion/ECM plasticity as a central adaptive axis in this radiation-adaptation model: cells curtail stable integrin-mediated anchorage to laminin and collagen, upshift fibronectin-driven matrix remodeling, and adopt a partial-EMT program that loosens epithelial identity without installing a full mesenchymal architecture. However, these EMT-associated changes should be interpreted cautiously in the context of radiation response. Recent work suggests that irradiation may not effectively reverse EMT and can instead promote a senescent drift that sustains cellular plasticity [55]. In our dataset, induction of CXCL8/IL-8 and MMP1 is consistent with pro-inflammatory, matrix-remodeling programs that could overlap with senescence biology [56,57]; however, we did not assay canonical senescence markers. Accordingly, future studies are needed to directly profile senescence programs alongside radiation treatment.

### 3.2. Mitochondrial Dysfunction and Compensatory Nuclear Responses

Mitochondrial control was reprogrammed at both the transcript and functional levels in MDA-MB-231^RR^ cells. RT-qPCR confirmed reduced expression of mtDNA-encoded respiratory genes (*MTCO1*, *MTCO1P12*, *MTND1*) alongside increased expression of nuclear-encoded regulators of mitochondrial transcription/translation (*MTERF4*, *MTRF1L*) (Figure 5). Consistent with impaired bioenergetic coupling, JC-1 imaging demonstrated a significant decrease in the J-aggregate/J-monomer ratio (Figure 6), indicating sustained depolarization of the inner membrane potential.

The RNA-seq dataset reinforces this picture of cellular respiratory stress with compensatory remodeling. Several electron transport chain (ETC) components and assembly factors were also reduced, including *MT-CO3*, *UQCRC1* (ubiquinol-cytochrome c reductase core protein 1), *ATP5MC1* (ATP synthase membrane subunit c), *MTIF3* (mitochondrial translation initiation factor 3), and *NDUFAF7* (NADH dehydrogenase complex assembly factor 7) (Appendix A) [58]. In parallel, selected nuclear subunits and import machinery were elevated, such as *COX7B* (cytochrome c oxidase subunit 7B), *NDUFA1* (NADH dehydrogenase subunit A1), and *TIMM8A* (translocase of inner mitochondrial membrane 8A), consistent with a compensatory attempt to stabilize electron-transport capacity and protein handling under stress [59]. Notably, an adenine nucleotide translocator isoform switch was evident, with *SLC25A5* (ANT2) increased and *SLC25A6* (ANT3) decreased, a configuration frequently associated with glycolytic, proliferation-biased states (Appendix A) [12,60]. These findings suggest that in the presence of decreased mitochondrial membrane potential, ATP supply is maintained by favoring ANT2-mediated ATP/ADP exchange, bolstered mitochondrial proteostasis through reinforced import and assembly modules, and reduced apoptotic priming while limiting ROS amplification [61]. Together, these features suggest a radiation-tolerant steady state that accommodates mitochondrial inefficiency and nominate ANT2 and mito-translation/import nodes as testable therapeutic liabilities.

At the metabolic level, these changes align with a Warburg-like shift. Upregulation of glycolytic enzymes such as *PGK1* (phosphoglycerate kinase 1) and *PGM2* (phosphoglucomutase 2) occurred alongside downregulation of multiple respiratory chain constituents, and with functional depolarization by JC-1 (Figure 5 and Figure 6; Appendix A) [62]. Such reprogramming would reduce reliance on tightly coupled oxidative phosphorylation, favor ATP generation via aerobic glycolysis, and increase flexibility under fluctuating oxygen and nutrient availability during fractionated radiation [63].

Mitochondrial depolarization also provides a credible trigger for retrograde signaling to the nucleus [64]. The transcriptome showed induction of stress-responsive mediators, including *ATF3* (activating transcription factor 3) and *EGR1* (early growth response 1), and anti-apoptotic effectors such as *BCL2A1* (Bcl-2-related protein A1) and *BIRC2* (baculoviral IAP repeat-containing protein 2) (Appendix A), consistent with reduced mitochondrial apoptotic priming [65]. Together with elevated *CASP4* (caspase 4) and *TNFAIP3* (TNF alpha induced protein 3), this pattern suggests a state that tolerates oxidative and inflammatory stress while limiting caspase engagement; features that would favor survival through repeated irradiation [66]. In this context, mitochondrial–nuclear signaling offers a mechanistic bridge from bioenergetic stress to the adhesion/phenotypic plasticity observed in this study, by modulating redox-sensitive transcription and cytoskeletal programs [67].

In sum, convergent transcriptional and functional evidence indicates that mitochondrial depolarization is a core feature of radiation-adapted cells, accompanied by selective down-tuning of cellular respiratory complexes, compensatory reinforcement of import/translation modules, a likely glycolytic shift, and transcriptional circuits that blunt apoptosis. This mitochondrial-centric reprogramming likely provides both a survival advantage under radiation stress and a plausible upstream influence on broader cellular plasticity.

### 3.3. Additional Networks Potentially Modulating Radiation Adaptation

To complement the validated adhesion and mitochondrial findings, the RNA-seq dataset suggested several additional, internally consistent signaling modules that may contribute to the molecular mechanisms underlying radiation adaptation. Since these gene networks are derived from transcriptional patterns without direct functional verification in this study, we present them as hypothesis-generating leads.

First, the RNA-seq profile pointed to a stress–inflammation module coupled to the cyclooxygenase-2/prostaglandin E2 (COX-2/PGE_2_) axis that plausibly links radiation exposure to survival signaling and adhesion remodeling. For example, immediate-early regulators of stress were increased, including *EGR1*, *ATF3*, and *GADD45A/B* (growth arrest and DNA damage inducible α/β), together with feedback dampeners *ZFP36* (zinc finger protein 36) and *TNFAIP3* (TNF-α–induced protein 3) (Appendix A) [68,69]. In parallel, *PTGS2* (prostaglandin-endoperoxide synthase 2/COX-2) was elevated, and receptor remodeling included *PTGER4* (prostaglandin E receptor 4) upregulation, a pattern compatible with PGE_2_→PTGER4 signaling into cAMP and PI3K–Akt pathways [70,71]. Within our dataset, this link was internally coherent: pro-remodeling and pro-survival outputs such as *FN1*, *MMP1*, *MMP3*, and *CXCL8* were increased, while the integrin repertoire (*ITGB1*, *ITGA2*, *ITGA6*) was reduced (Appendix A), and the JC-1 assay showed a sub-catastrophic decrease in mitochondrial membrane potential. Taken together, a radiation-evoked stress program coupled to COX-2/PGE_2_ signaling could buffer apoptotic execution and promote adhesion/ECM plasticity in MDA-MB-231^RR^ cells [72].

Second, the RNA-seq profile pointed to a recalibration of the inflammation–immune-visibility axis that could favor survival while limiting immune detection [73]. For example, pathway enrichment for complement/coagulation and cytokine signaling (Table 2 and Table 3) coincided with increased *CD274* (PD-L1) and *CXCL8*, alongside reduced expression of multiple HLA class I/II transcripts and *STAT1* (Appendix A) [74]. In parallel, components of the hemostatic interface, *F3* (tissue factor) and *PROS1* (protein S), were elevated (Appendix A) [75,76]. In our model of radiation-adapted TNBC cells, this combination is relevant because PD-L1 upregulation can blunt effector T-cell cytotoxicity during the post-irradiation window [77], while lower HLA/STAT1 may dampen interferon-driven antigen presentation and immunogenic cell death [78]. Furthermore, elevated CXCL8 can bias toward neutrophil recruitment and pro-angiogenic signaling that stabilizes damaged vasculature [79], and higher F3/PROS1 suggests a thrombo-inflammatory niche (fibrin deposition, microthrombi) that physically shields tumor cells and may facilitate intravasation [80]. Together, the transcriptomic pattern hints toward an inflammatory, clot-prone microenvironment that buffers stress signals and supports survival after repeated radiation [81].

Third, the RNA-seq dataset identified several transcriptional control nodes that may contribute to adaptation to repeated radiation. For example, nuclear receptors and dormancy-linked factors were increased, including *NR2F1* (nuclear receptor subfamily 2 group F member 1), *RORA* (RAR-related orphan receptor A), and *RORB* (RAR-related orphan receptor B), suggesting a shift toward stress-tolerant, slow-cycling programs (Appendix A) [82,83]. In addition, developmental and EMT-adjacent regulators also increased, such as *KLF8* (Kruppel-like factor 8), *HEYL* (HEY-like transcription factor), *MSX2* (msh homeobox 2), *PRDM1* (PR/SET domain 1), *ID4* (inhibitor of DNA binding 4), *LMO2* (LIM domain only 2), and *MITF* (microphthalmia-associated transcription factor) [84]. In parallel, components that restrain Hippo signaling were altered, with *VGLL3* (vestigial-like family member 3, a TEAD cofactor) increased and *LATS2* (large tumor suppressor kinase 2) decreased, a configuration that can favor TEAD-dependent survival and matrix-remodeling outputs [85,86]. Conversely, several nodes linked to interferon signaling, lineage specification, and chromatin constraint were reduced, including *STAT1* [87], *ELF1* (E74-like ETS transcription factor 1) [88], *RUNX2* (runt-related transcription factor 2) [89], *HDAC5* (histone deacetylase 5) [37], and *CBX8* (chromobox 8) (Appendix A) [90]. Taken together, the directionality of these changes is consistent with radiation-evoked reprogramming toward transcriptional states that support stress endurance, matrix remodeling, and reduced immune visibility, while deprioritizing interferon-driven surveillance and rigid lineage enforcement.

### 3.4. Limitations and Future Directions

This study has several limitations. First, all findings were generated using a single TNBC cell line (MDA-MB-231). Given the well-recognized heterogeneity of TNBC, these results may not fully generalize to other molecular subtypes or patient-derived breast cancer tumors. Future studies incorporating additional TNBC lines and patient-derived models will be needed to determine which transcriptional and functional adaptations are broadly conserved. Second, our repeated-radiation model involved clonal selection following chronic irradiation. While this approach provides a stable and well-defined radiation-adapted population, it may also enrich for rare pre-existing resistant clones, rather than solely representing de novo adaptations that occur during routine clinical treatment. These factors should be considered when extrapolating our findings to patient tumors and treatment settings. Nonetheless, the results presented in this study provide valuable insight into the molecular and functional consequences of long-term radiation exposure in TNBC and establish a foundation for future studies aimed at overcoming radiation resistance.

## 4. Methods

### 4.1. Cell Culture

MDA-MB-231 cells, purchased from American Type Culture Collection (Cat. #: HTB-26, Cedarlane Labs, Burlington, ON, Canada), were maintained in Dulbecco’s Modified Eagle’s Media (DMEM; Cytiva HyClone™, Logan, UT, USA, Cat. #: SH3028501) supplemented with 10% fetal bovine serum (Cytiva HyClone™, Cat. #: SH3039603HI), 1 U/mL penicillin–streptomycin solution (Cytiva HyClone™, Cat. #:SV30010), and 6 mM L-glutamine (Cytiva HyClone™, Cat. #: SH3003401). Cells were incubated at 37 °C with humid air and 5% CO_2_. Cells were maintained in T-75 flasks and either plated into T-25 flasks for RNA extraction or plated into 100 mm dishes for protein extraction.

### 4.2. Development of MDA-MB-231^RR^ Cells

MDA–MB-231 cells were cultured in T-75 flasks and subjected to a fractionated radiation regimen modeled after a previously established protocol for generating radiation-resistant breast cancer cell lines [91]. Cells were irradiated once weekly using an X-RAD 320 irradiation cabinet (Precision X-Ray, Madison, CT, USA) operating at 320 kV and 12.5 mA with a 2 mm aluminum filter, delivering doses at a rate of 1.5 Gy/min. The initial dose was 2 Gy, with weekly increments of 0.5 Gy until a final dose of 7.5 Gy was reached in week 11, resulting in a cumulative dose of 57 Gy over the full course of treatment.

Following this fractionated radiation regimen, surviving cells were subjected to clonal isolation using the Limiting Dilution method as previously described [92]. Cells were diluted to achieve an approximate concentration of one cell per mL and plated across multiple 12-well plates. After 12 h, wells were visually inspected to confirm the presence of a single adherent cell, which was then monitored for clonal expansion. Approximately 10 clones demonstrated initial outgrowth, and of these, four clones successfully proliferated and were expanded to establish stable clonal cell lines. These radiation-adapted clones, collectively referred to as MDA-MB-231^RR^ (Repeated Radiation), were used in all downstream experiments. All downstream functional and transcriptomic assays were performed within 4–6 passages after clonal expansion to ensure retention of the radiation-adapted phenotype.

In parallel, control MDA-MB-231 cells were subjected to identical culture conditions and passaging schedules, but without exposure to radiation. These sham-matched controls (MDA-MB-231^Control^) were maintained in parallel and served as the baseline comparator in all subsequent analyses.

### 4.3. RNA Sample Preparation

Cells were plated into the necessary number of T-25 flasks and allowed to grow to 70% confluency before the media were removed, the adhered cells were washed with ice cold PBS, and 1 mL of TRIzol was added per flask. Once all cells were lysed by the TRIzol, the solution was transferred to a microcentrifuge tube for a standard TRIzol extraction [15]. Purified RNA concentration was determined via a NanoDrop spectrophotometer (ThermoScientific, Waltham, MA, USA, Catalog #: ND-1000).

### 4.4. RNA Sequencing and Gene Ontology

RNA samples were further purified via the NEB Monarch^®^ RNA cleanup kit (NEB, Ipswich, MA, USA, Cat. #: T2040L). The NEBNext^®^ Ultra™ II Directional RNA Library Kit (NEB, Ipswich, MA, USA, Cat. #: E7765S) was then used for library preparation. This kit isolated the poly-A messenger ribonucleic acid (mRNA) and created strand-specific directional libraries using the deoxyuridine triphosphate (dUTP) method. The prepared library samples were then quantified using the NEBNext^®^ Library Quant Kit (NEB, Ipswich, MA, USA, Cat. #: E7630S). These samples were combined at an equimolar concentration and sequenced at The Centre for Applied Genomics (TCAG, SickKids, Toronto, ON, Canada) using a NovaSeq 6000 Illumina platform, achieving roughly 35 million reads per sample (paired-end 100 base-pair reads). The sequencing data was processed in-house using a variety of bioinformatics toolkits provided by DRAGEN Inc. through the Illumina Sequence Hub platform (Illumina, San Diego, CA, USA) [15,93]. Specifically, *DRAGEN FASTQ* was utilized for sequencing data quality checks and read trims, while *DRAGEN RNA* was used to align reads to the human reference genome (GRCh38.p13) and carry out transcript-count analysis. DRAGEN Differential Expression was used to analyze differential expression based on the DESeq2 platform. The criteria for selecting differentially expressed genes were set based on a gene-fold change of less than −2 or greater than 2, false discovery rate (FDR) corrected *p*-values of less than 0.05, and a minimum average read count of 40 transcripts per million (TPM) Lastly, iPathwayGuide (AdvaitaBio, Ann Arbor, MI, USA) was used to perform gene ontology analysis as described previously [94,95,96].

### 4.5. cDNA Synthesis and RT-qPCR

Complementary DNA (cDNA) synthesis was performed using the cDNA SuperMix Synthesis Kit (Medi-Res Corp., Sudbury, ON, Canada; Cat. #: Bi2M-SSRT3) according to the manufacturer’s instructions, with minor modifications. Briefly, 2 μg of total RNA was treated with DNase and reverse-transcribed using random hexamers and the provided reverse transcriptase enzyme. The initial reaction volume was set to 20 μL and subsequently adjusted to 50 μL to obtain a final cDNA concentration of 40 ng/μL. cDNA samples were stored at −80 °C until use.

Forward and reverse primer pairs for SYBR Green-based RT-qPCR analysis were designed in-house and validated under standard conditions (efficiency between 90–110% and R^2^ > 0.99) [97,98]. Validated primer sequences used for RT-qPCR are summarized in Table 3. RT-qPCR reactions were performed in 96-well plates using the QuantStudio 5 Real-Time PCR instrument (ThermoScientific, Waltham, MA, USA; Cat. #: A34322). Each 15 μL reaction contained 10 ng of cDNA, 600 nM of both forward and reverse primers, and 1× SYBR qPCR Master Mix (Medi-Res Corp., Cat. #: Bi2M-2xqPCR). The cycling protocol consisted of an initial denaturation at 95 °C for 15 s, followed by 40 amplification cycles of 95 °C for 15 s and 60 °C for 30 s. Fluorescence measurements were taken at the end of each extension phase.

Amplification specificity was assessed by melt curve analysis following the final PCR cycle. The fluorescence data were analyzed using QuantStudio™ Design and Analysis Software v1.5.1 (Applied Biosystems, Foster City, USA) to determine quantification cycle (Cq) values. Gene expression levels were normalized to the geometric mean of three reference housekeeping genes: *HSPD1*, *TUBA1B*, and *ATP5F1B*. Relative mRNA expression was calculated using the 2^−ΔΔCq^ method [99], and values are reported as mean fold change ± standard error of the mean (SEM).

### 4.6. Cell Adhesion Assay

Cell adhesion assay was performed as described previously with minor modifications [15]. Briefly, 7.5 ng/μL fibronectin (Sigma-Aldrich, St. Louis, MO, USA, Cat. #: F2006), 20 ng/μL laminin (Sigma-Aldrich, Cat. #: L2020), 25 ng/μL collagen (Sigma-Aldrich, Cat. #: C5533), or 100 ng/μL poly-d-lysine positive control (Sigma-Aldrich, Cat. #: P7280) was spread on flat-bottom 96-well plates and left to incubate at 4 °C overnight. The plates were then rinsed with a wash solution (0.2% BSA in PBS) and blocked with 2% BSA. The cells were gathered by trypsinization and resuspended in cell recovery media (serum-free media supplemented with 20 mM HEPES and 0.1% bovine serum albumin, pH 7.4). After a 30 min cell recovery period at 37 °C, the cells were plated at 40,000 cells/well. After one hour, the cells spread on fibronectin, laminin and collagen were rinsed three times with a prewarmed wash solution. All wells were then fixed with 4% paraformaldehyde for 10 min at room temperature, followed by a 0.1% crystal violet stain for 30 min at room temperature. The wells were rinsed three times with autoclaved deionized water (200 μL/wash), aspirated, and air-dried. Brightfield images were acquired using the BioTek Cytation 5 imager (Agilent Technologies, Winooski, Vermont, USA) at 4× magnification. Cell quantification was performed using FIJI (ImageJ 2.14.0, Washington, DC, USA). Images were converted to 8-bit, background subtracted (50-pixel radius, light background), and threshold was set using the auto setting. Watershed separation was applied prior to particle analysis. For poly-D-lysine–coated wells, particle size was set to 25–300 pixels (circularity 0.25–1.00), and for ECM-coated wells, 75–300 pixels. Cell counts from fibronectin, laminin, and collagen-coated wells were normalized to poly-D-lysine controls. Relative adhesion is expressed as fold change ± SEM relative to MDA-MB-231^Control^.

### 4.7. Mitochondrial Membrane Potential Assay

Mitochondrial membrane potential was assessed using JC-1 dye (Invitrogen™, Thermo Fisher Scientific, Waltham, MA, USA; Cat. #: T3168) according to the manufacturer’s protocol with minor modifications. MDA-MB-231^Control^ and MDA-MB-231^RR^ cells were seeded into poly-D-lysine-coated black 96-well plates at densities of 20,000–25,000 cells/well and incubated overnight at 37 °C in a 5% CO_2_ humidified incubator. The next day, cells were washed once with PBS and incubated with 100 µL of a 20 µM JC-1 working solution per well for 30 min at 37 °C in the dark. A DMSO-only control was included to account for potential solvent effects. Following incubation, cells were washed twice with PBS to remove excess dye, and 100 µL of fresh PBS was added to each well. Imaging was performed immediately using a BioTek Cytation 5 Cell Imaging Multimode Reader. Fluorescence was measured using standard filter sets to detect JC-1 monomers (green fluorescence; Excitation/Emission = 485/530 nm) and J-aggregates (red fluorescence; Excitation/Emission = 540/590 nm). Mitochondrial membrane potential was quantified by calculating the red-to-green fluorescence intensity ratio on a per-cell basis using ImageJ. Fluorescence measurements were normalized to the surface area of individual cells. For each well, 8 random fields were imaged, and 20–30 cells were analyzed per image. Three wells were assessed per condition in each experiment, and the assay was repeated across four independent biological replicates. Data are presented as the mean red/green fluorescence ratio ± standard error of the mean (SEM) averaged across the four experiments.

### 4.8. Statistical Analysis

Statistical analyses were performed using both bioinformatics pipelines and conventional statistical software. For RNA-seq data, differential gene expression was calculated using the DRAGEN Differential Expression pipeline based on the DESeq2 framework, which models read counts using a negative binomial distribution as described previously [100]. Significance thresholds were set at a fold change >2 or <−2, a false discovery rate (FDR)-adjusted *p*-value < 0.05, and a minimum average transcript abundance of 40 TPM. Gene ontology (GO) and pathway enrichment analyses were performed using iPathwayGuide (AdvaitaBio) with FDR correction for multiple comparisons as shown previously [94,95].

For functional assays (RT-qPCR, cell adhesion, mitochondrial membrane potential), data were analyzed in Jamovi (v2.5) using one-way ANOVA followed by Tukey’s post hoc test for multiple group comparisons or unpaired two-tailed *t*-tests for two-group comparisons, as appropriate. Data are presented as mean ± SEM, and statistical significance was set at *p* < 0.05.

## 5. Conclusions

This study defines key molecular and functional adaptations that arise in TNBC cells following repeated radiation exposure. Transcriptomic profiling combined with functional assays revealed convergent alterations in three principal domains: (1) remodeling of adhesion and extracellular matrix interactions, characterized by reduced integrin repertoire and impaired substrate attachment; (2) mitochondrial dysfunction, reflected by diminished membrane potential without overt collapse of viability; and (3) features of partial mesenchymal transition, including induction of *SNAI2* and *FN1* alongside loss of epithelial markers, but without full acquisition of a mesenchymal program. Collectively, these adaptations indicate that radiation pressure drives a transcriptional and phenotypic reprogramming toward a more plastic, stress-tolerant cellular state in which survival is supported despite compromised energetics, weakened adhesion, and incomplete lineage switching.

From a therapeutic standpoint, these adaptations highlight actionable vulnerabilities in radiation-resistant TNBC. Strategies that destabilize adhesion signaling (e.g., integrin or focal adhesion kinase inhibitors) or exacerbate mitochondrial dysfunction (e.g., metabolic inhibitors targeting oxidative phosphorylation) may selectively impair survival of radiation-adapted cells. In addition, transcriptomic signatures pointed to auxiliary pathways (COX-2/prostaglandin signaling, immune checkpoint regulation, and pro-thrombotic microenvironmental remodeling) that could further reinforce resistance. These modules warrant future exploration as complementary therapeutic targets to counteract radiation adaptation.

Collectively, our findings provide a systems-level view of the molecular rewiring associated with radiation resistance in TNBC and nominate candidate pathways for rational therapeutic intervention. By focusing on adhesion and mitochondrial vulnerabilities, and by probing additional signaling axes that emerge under radiation pressure, this work lays a foundation for translational strategies aimed at improving local control and long-term outcomes in patients with TNBC undergoing radiotherapy.

## Figures and Tables

**Figure 1 ijms-26-09611-f001:**
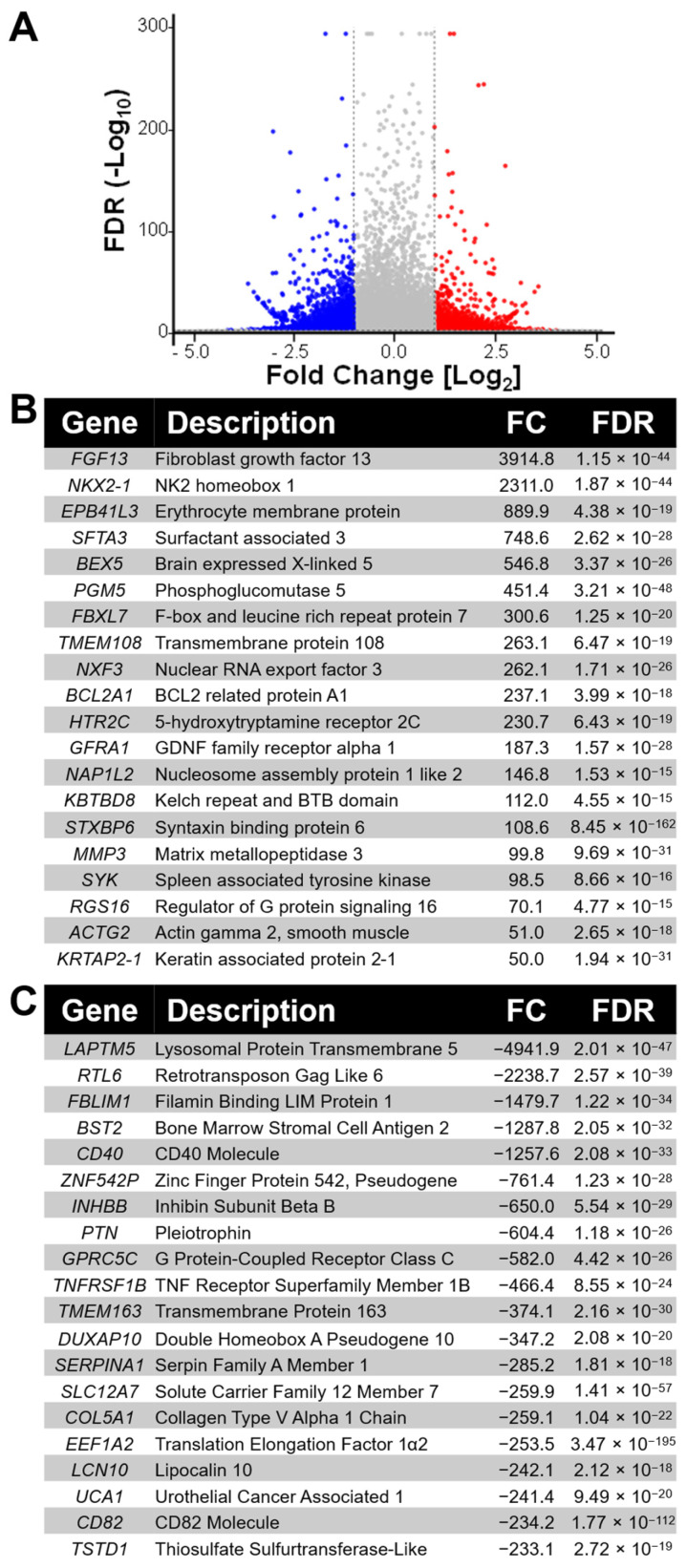
**RNA-sequencing reveals global transcriptional reprogramming in radiation-adapted MDA-MB-231^RR^ cells.** (**A**) Volcano plot illustrating DEGs in MDA-MB-231^RR^ cells relative to MDA-MB-231^Control^ cells. A total of 13,590 mRNA transcripts were detected by whole-transcriptome RNA-sequencing, of which 1572 DEGs met the significance criteria of fold change (FC) >2 or <−2, FDR-adjusted *p*-value < 0.05, and minimum average transcript abundance of 40 TPM. Red and blue circles represent significantly upregulated and downregulated DEGs, respectively; grey circles denote non-significant changes. The x-axis displays log_2_ fold change, while the y-axis shows the −log_10_ FDR. The outer edges of the plot indicate the most strongly dysregulated genes, and genes higher on the plot show greater statistical significance. (**B**) Top 20 upregulated genes and (**C**) top 20 downregulated genes in MDA-MB-231^RR^ cells relative to MDA-MB-231^Control^ cells, ranked by fold change. Gene descriptions, fold changes, and FDR-adjusted *p*-values are included.

**Figure 2 ijms-26-09611-f002:**
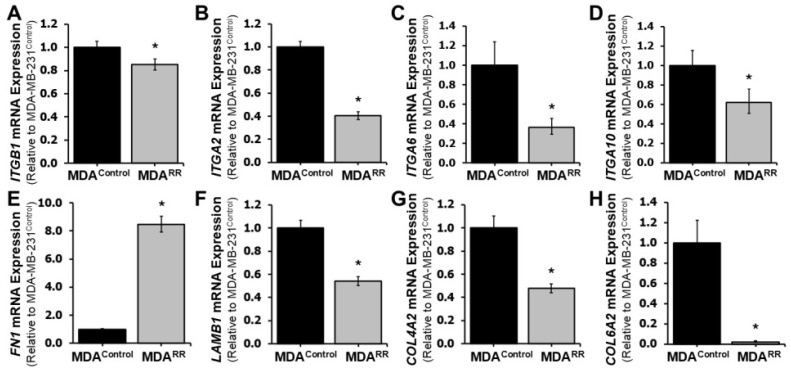
**Radiation-adapted MDA-MB-231^RR^ cells exhibit altered expression of adhesion-related genes.** RT-qPCR analysis of key genes involved in cell–ECM adhesion comparing MDA-MB-231^RR^ cells (MDA^RR^) to MDA-MB-231^Control^ (MDA^Control^) cells. Data are expressed relative to MDA-MB-231^Control^ and represent mean ± SEM. In MDA-MB-231^RR^ cells, expression of integrin subunits *ITGB1* (**A**), *ITGA2* (**B**), *ITGA6* (**C**), and *ITGA10* (**D**) was significantly reduced. Expression of the ECM components *FN1* (**E**) was significantly increased, while expression of *LAMB1* (**F**), *COL4A2* (**G**) and *COL6A2* (**H**) were significantly downregulated. Asterisks indicate statistical significance (* *p* < 0.05). Abbreviations: *ITGB1* (Integrin Subunit Beta 1), *ITGA2* (Integrin Subunit Alpha 2), *ITGA6* (Integrin Subunit Alpha 6), *ITGA10* (Integrin Subunit Alpha 10), *FN1* (Fibronectin 1), *LAMB1* (Laminin Subunit Beta 1), *COL4A2* (Collagen Type IV Alpha 2 Chain), *COL6A2* (Collagen Type VI Alpha 2 Chain).

**Figure 3 ijms-26-09611-f003:**
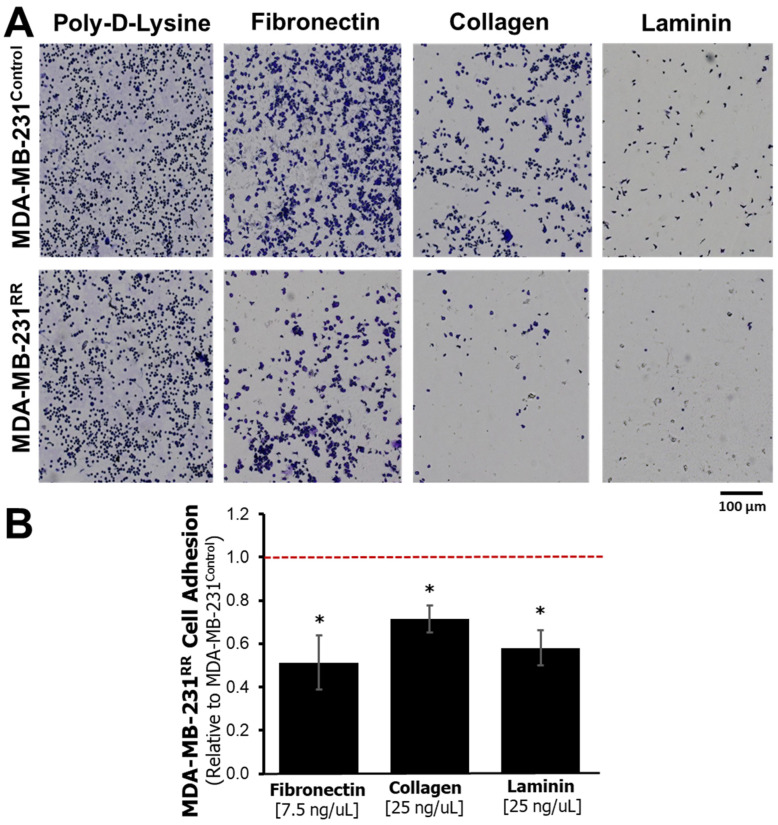
**Cell adhesion is reduced in radiation-adapted MDA-MB-231^RR^ cells.** (**A**) Representative images of crystal violet-stained MDA-MB-231^Control^ and MDA-MB-231^RR^ cells seeded on wells coated with poly-D-lysine (control), fibronectin (7.5 ng/μL), collagen (25 ng/μL), or laminin (25 ng/μL). MDA-MB-231^RR^ cells showed visibly reduced adhesion to fibronectin, collagen, and laminin compared to parental controls. Scale bar = 100 μm. (**B**) Quantification of adhered MDA-MB-231^RR^ cells relative to MDA-MB-231^Control^ cells on each ECM substrate. The red dotted line represents the control adhesion level (set to 1.0). Data are shown as mean ± SEM (*n* = 4). Adhesion ratios for fibronectin, collagen, and laminin were 0.410 (*p* = 0.032), 0.574 (*p* = 0.032), and 0.211 (*p* = 0.021), respectively (*, *p* < 0.05).

**Figure 4 ijms-26-09611-f004:**
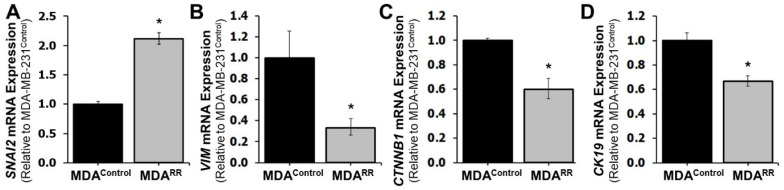
Radiation-adapted MDA-MB-231^RR^ cells exhibit transcriptional changes consistent with an epithelial-to-mesenchymal transition (EMT) phenotype. The mRNA expression levels of key EMT-related genes were quantified by RT-qPCR in MDA-MB-231^RR^ cells (MDA^RR^) and MDA-MB-231^Control^ (MDA^Control^) cells. (**A**–**D**) show significantly altered expression of *SNAI2* (**A**), *VIM* (**B**), *CTNNB1* (**C**), and *CK19* (**D**) in radiation-adapted cells relative to controls. Additional EMT genes (*ACTA2*, *AKT2*, *CD44*, *CDH2*, and *SNAI1*) were profiled but did not show significant differences between the cell lines. Bars represent mean ± SEM. Statistical significance was determined using an unpaired two-tailed *t*-test (*, *p* < 0.05). Abbreviations: *SNAI2* (Snail Family Transcriptional Repressor 2), *VIM* (Vimentin), *CTNNB1* (Catenin Beta 1), *CK19* (Cytokeratin 19).

**Figure 5 ijms-26-09611-f005:**
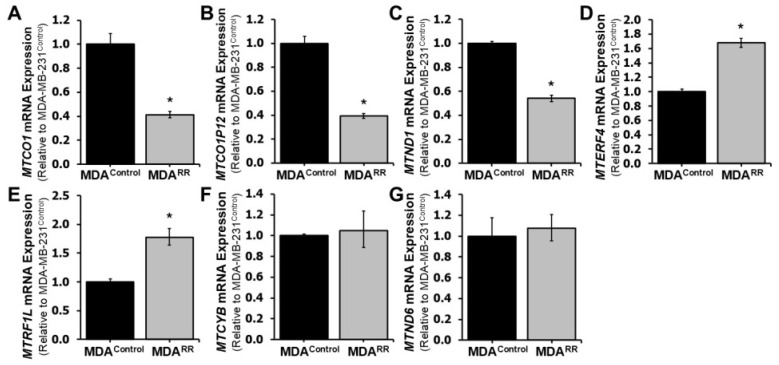
**Mitochondrial gene expression is altered in MDA-MB-231^RR^ cells.** RT-qPCR analysis was performed to assess the expression of key mitochondrial genes in MDA-MB-231^RR^ cells (MDA^RR^) relative to MDA-MB-231^Control^ (MDA^Control^) cells. Mitochondrial-encoded genes *MTCO1* (**A**), *MTCO1P12* (**B**), and *MTND1* (**C**) were significantly downregulated in MDA-MB-231^RR^ cells. (**D**) Expression of the nuclear-encoded mitochondrial gene *MTERF4* was significantly upregulated in MDA-MB-231^RR^ cells. (**E**) *MTRF1L* expression was also significantly increased in MDA-MB-231^RR^ cells. Expression of MTCYB (**F**) and MTND6 (**G**) was not significantly different between MDA-MB-231^RR^ and MDA-MB-231^Control^ cells. Gene expression levels are shown relative to MDA-MB-231^Control^ and represent mean ± SEM. Statistical significance is indicated by asterisks (* *p* < 0.05). Abbreviations: *MTCO1* (Cytochrome C Oxidase Subunit I), *MTCO1P12* (MT-CO1 Pseudogene 12), *MTND1* (NADH Dehydrogenase Subunit 1), *MTERF4* (Mitochondrial Transcription Termination Factor 4), *MTRF1L* (Mitochondrial Translational Release Factor 1-Like), *MTCYB* (Cytochrome B), *MTND6* (NADH Dehydrogenase Subunit 6).

**Figure 6 ijms-26-09611-f006:**
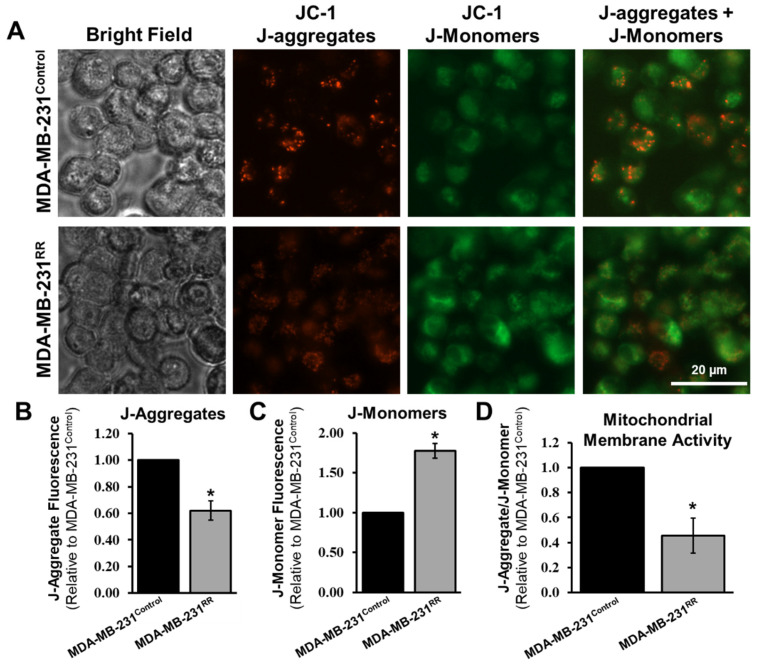
**Mitochondrial membrane potential is reduced in MDA-MB-231^RR^ cells.** Representative images of MDA-MB-231^Control^ and MDA-MB-231^RR^ cells stained with JC-1 dye are shown in panel (**A**), captured using Cytation5 under brightfield, red fluorescence (J-aggregates), green fluorescence (J-monomers), and merged channels. Scale bar = 20 µm. Quantification of fluorescence intensities revealed a significant reduction in J-aggregates (**B**) and a significant increase in J-monomers (**C**) in MDA-MB-231^RR^ cells relative to MDA-MB-231^Control^, indicating a shift toward mitochondrial depolarization. The resulting ratio of J-aggregates to J-monomers (**D**), which reflects mitochondrial membrane potential, was significantly decreased by approximately 50% in MDA-MB-231^RR^ cells. Data represent mean ± SEM from at least three independent experiments, with statistical significance denoted by asterisks (* *p* < 0.05).

**Table 1 ijms-26-09611-t001:** Gene ontology (GO) enrichment analysis of differentially expressed genes in MDA-MB-231^RR^ cells relative to MDA-MB-231^Control^ cells. The top 10 significantly enriched GO biological processes are shown, ranked by FDR-adjusted *p*-value. The table includes the GO term ID, process name, number of DEGs associated with each term, the total number of genes annotated to each GO term in the reference database (Total Genes), and the corresponding FDR. GO analysis was performed using iPathwayGuide (AdvaitaBio) based on DEGs identified from RNA-seq analysis of MDA-MB-231^RR^ cells compared to MDA-MB-231^Control^ cells. Enriched GO terms highlight biological processes associated with radiation adaptation, including altered cell adhesion, signaling pathways, and regulation of multicellular organization.

Gene Ontology ID	Biological Process	DEGs	Total Genes	FDR
GO:0022610	Biological Adhesion	214	805	3.81 × 10^−18^
GO:0007155	Cell Adhesion	212	802	6.50 × 10^−18^
GO:0032501	Multicellular Organismal Process	722	3968	4.62 × 10^−16^
GO:0051239	Regulation of Multicellular Organismal Process	323	1515	4.11 × 10^−13^
GO:0009653	Anatomical Structure Morphogenesis	336	1621	4.91 × 10^−12^
GO:0098609	Cell–Cell Adhesion	130	468	4.91 × 10^−12^
GO:0023052	Signaling	627	3509	5.32 × 10^−11^
GO:0007154	Cell Communication	632	3552	8.66 × 10^−11^
GO:0007165	Signal Transduction	580	5258	2.02 × 10^−9^
GO:0050896	Response to Stimulus	830	4973	3.12 × 10^−9^

**Table 2 ijms-26-09611-t002:** Top 10 enriched signaling pathways associated with radiation adaptation in MDA-MB-231^RR^ cells. DEGs in MDA-MB-231^RR^ cells relative to MDA-MB-231^Control^ cells were analyzed using iPathwayGuide (AdvaitaBio) to identify significantly enriched KEGG signaling pathways. The table lists the top 10 pathways ranked by FDR-adjusted *p*-values, along with their KEGG pathway identifiers, and the upregulated and downregulated DEGs associated with each pathway. These findings highlight key signaling networks altered during radiation adaptation, including those involved in inflammation, extracellular matrix interactions, cell adhesion, and cytokine signaling.

Pathway (ID)	FDR	Upregulated DEGs	Downregulated DEGs
Complement and Coagulation Cascades (04610)	0.008	*FGB*, *F2RL2*, *SERPINB2*, *F3*, *CD55*, *ITGB2*, *PROS1*	*SERPINA1*, *CFD*, *C1R*, *PLAU*, *C4BPB*, *F12*
Neuroactive Ligand-Receptor Interaction (04080)	0.010	*HTR2C*, *GABRA3*, *P2RY10*, *GABBR2*, *F2RL2*, *ADM*, *S1PR3*, *GRPR*, *PRLR*, *KISS1*, *PTGER4*, *HRH1*	*VIPR1*, *SSTR2*, *NMB*, *GRIK4*, *ADORA2A*, *S1PR5*, *PTGER2*, *HTR7*, *P2RY2*, *LPAR2*
Cell Adhesion Molecules (04514)	0.027	*VCAN*, *NEO1*, *CD22*, *CD274*, *ITGB2*, *CDH4*	*CD40*, *L1CAM*, *CLDN2*, *ICAM1*, *HLA-DRB1*, *ITGA6*, *NRCAM*, *HLA-A*, *HLA-DRA*, *CLDN3*, *NCAM2*, *VSIR*, *HLA-F*, *ICOSLG*, *HLA-C*, *HLA-B*
ABC Transporters (02010)	0.033	*ABCB7*, *ABCA3*	*ABCC3*, *ABCB9*, *ABCG2*, *TAP1*, *ABCA2*, *ABCD1*, *TAP2*, *ABCA1*, *ABCC4*
Pathways in Cancer (05200)	0.033	*CCND2*, *MMP1*, *PTGS2*, *LAMA1*, *HEYL*, *FN1*, *PLCB1*, *IL7R*, *FGF5*, *COL4A5*, *FGF1*, *ESR2*, *PTGER4*, *IL13RA1*, *CAMK2D*, *HHIP*, *MGST3*, *WNT7B*, *MITF*, *EGF*, *GADD45B*, *GNAS*, *GADD45A*, *CXCL8*	*PDGFRB*, *EGLN3*, *FGFR4*, *JAG2*, *FHH*, *GSTM2*, *SUFU*, *JUP*, *PTGER2*, *PGF*, *FLT3LG*, *ITGA6*, *LAMA5*, *TERT*, *IL15RA*, *CDKN1A*, *ADCY6*, *FRAT1*, *NCOA1*, *TGFBR2*, *CCND3*, *IL15*, *ADCY7*, *CCNA1*, *PLCG1*, *DDB2*, *PLCG2*, *NCOA3*, *ITGA2*, *GSTM4*, *HES1*, *COL4A1*, *COL4A2*, *GNG11*, *RALB*, *PLD2*, *DVL2*, *LPAR2*, *LRP5*, *FZD1*, *TRAF5*, *EML4*, *TRAF3*, *AKT1*, *LAMB1*, *EPOR*, *STAT1*
Protein Digestion and Absorption (04974)	0.033	*CPA3*, *ATP1A3*, *COL8A1*, *COL4A5*	*COL5A1*, *COL6A2*, *COL6A3*, *COL27A1*, *COL7A1*, *COL13A1*, *COL4A1*, *COL4A2*, *KCNN4*
ECM-Receptor Interaction (04512)	0.033	*LAMA1*, *FN1*, *TNC*, *COL4A5*	*COL6A2*, *COL6A3*, *ITGB4*, *ITGA6*, *LAMA5*, *FREM2*, *ITGA2*, *COL4A1*, *COL4A2*, *ITGA10*, *DAG1*, *LAMB1*
AGE-RAGE Signaling Pathway in Diabetic Complications (04933)	0.033	*FN1*, *PLCB1*, *COL4A5*, *F3*, *PRKCZ*, *EGR1*, *CXCL8*	*NFATC1*, *ICAM1*, *PLCD1*, *MAPK13*, *TGFBR2*, *PLCG1*, *PLCG2*, *COL4A2*, *PRKCE*, *PLCD3*, *AKT1*, *STAT1*
Cytokine-Cytokine Receptor Interaction (04060)	0.034	*IL7R*, *TNFRSF10D*, *PRLR*, *IL13RA1*, *IL1RL2*, *CXCL8*	*CD40*, *INHBB*, *TNFRSF1B*, *NGFR*, *CX3CL1*, *IL11*, *GDF5*, *IFNLR1*, *TNFSF12*, *IL24*, *TNFRSF11B*, *IL15RA*, *CRLF2*, *TGFBR2*, *IL15*, *IL17RE*, *TNFRSF19*, *EPOR*
Insulin Secretion (04911)	0.040	*RYR2*, *ATP1A3*, *SNAP25*, *PLCB1*, *CAMK2D*, *KCNMB4*, *RIMS2*, *GNAS*	*KCNMB3*, *ADCY6*, *ADCY7*, *PCLO*, *ATF6B*, *KCNN4*

Abbreviations: *ABCA1* (ATP binding cassette subfamily A member 1), *ABCA2* (ATP binding cassette subfamily A member 2), *ABCA3* (ATP binding cassette subfamily A member 3), *ABCB7* (ATP binding cassette subfamily B member 7), *ABCB9* (ATP binding cassette subfamily B member 9), *ABCC3* (ATP binding cassette subfamily C member 3), *ABCC4* (ATP binding cassette subfamily C member 4 (PEL blood group)), *ABCD1* (ATP binding cassette subfamily D member 1), *ABCG2* (ATP binding cassette subfamily G member 2 (JR blood group)), *ADCY6* (adenylate cyclase 6), *ADCY7* (adenylate cyclase 7), *ADM* (adrenomedullin), *ADORA2A* (adenosine A2a receptor), *AKT1* (AKT serine/threonine kinase 1), *ATF6B* (activating transcription factor 6 beta), *ATP1A3* (ATPase Na+/K+ transporting subunit alpha 3), *C1R* (complement C1r), *C4BPB* (complement component 4 binding protein beta), *CAMK2D* (calcium/calmodulin dependent protein kinase II delta), *FHH* (calcium sensing receptor), *CCNA1* (cyclin A1), *CCND2* (cyclin D2), *CCND3* (cyclin D3), *CD22* (CD22 molecule), *CD274* (CD274 molecule), *CD40* (CD40 molecule), *CD55* (CD55 molecule (Cromer blood group)), *CDH4* (cadherin 4), *CDKN1A* (cysteine rich DPF motif domain containing 1), *CLDN2* (claudin 2), *CLDN3* (claudin 3), *F3* (contactin 1), *COL13A1* (collagen type XIII alpha 1 chain), *COL27A1* (collagen type XXVII alpha 1 chain), *COL4A1* (collagen type IV alpha 1 chain), *COL4A2* (collagen type IV alpha 2 chain), *COL4A5* (collagen type IV alpha 5 chain), *COL5A1* (collagen type V alpha 1 chain), *COL6A2* (collagen type VI alpha 2 chain), *COL6A3* (collagen type VI alpha 3 chain), *COL7A1* (collagen type VII alpha 1 chain), *COL8A1* (collagen type VIII alpha 1 chain), *CPA3* (carboxypeptidase A4), *CRLF2* (cytokine receptor like factor 2), *CX3CL1* (C-X3-C motif chemokine ligand 1), *CXCL8* (C-X-C motif chemokine ligand 8), *DAG1* (dystroglycan 1), *DDB2* (damage specific DNA binding protein 2), *HRH1* (DExH-box helicase 34), *DVL2* (dishevelled segment polarity protein 2), *EGF* (epidermal growth factor), *EGLN3* (egl-9 family hypoxia inducible factor 3), *EGR1* (early growth response 1), *EML4* (EMAP like 4), *EPOR* (erythropoietin receptor), *ESR2* (estrogen receptor 2), *F12* (coagulation factor XII), *F2RL2* (coagulation factor II thrombin receptor like 2), *FGB* (fibrinogen beta chain), *FGF1* (fibroblast growth factor 1), *FGF5* (fibroblast growth factor 5), *FGFR4* (fibroblast growth factor receptor 4), *FLT3LG* (fms related receptor tyrosine kinase 3 ligand), *FN1* (fibronectin 1), *FRAT1* (FRAT regulator of Wnt signaling pathway 1), *FREM2* (FRAS1 related extracellular matrix 2), *FZD1* (frizzled class receptor 1), *GABBR2* (gamma-aminobutyric acid type B receptor subunit 2), *GABRA3* (gamma-aminobutyric acid type A receptor subunit alpha3), *GADD45A* (growth arrest and DNA damage inducible alpha), *GADD45B* (growth arrest and DNA damage inducible beta), *HES1* (glutamine amidotransferase class 1 domain containing 3), *GDF5* (growth differentiation factor 5), *GNAS* (GNAS complex locus), *GNG11* (G protein subunit gamma 11), *NMB* (glycoprotein nmb), *GRIK4* (glutamate ionotropic receptor kainate type subunit 4), *GRPR* (gastrin releasing peptide receptor), *GSTM2* (glutathione S-transferase mu 2), *GSTM4* (glutathione S-transferase mu 4), *HEYL* (hes related family bHLH transcription factor with YRPW motif like), *HHIP* (hedgehog interacting protein), *HLA-A* (major histocompatibility complex, class I, A), *HLA-B* (major histocompatibility complex, class I, B), *HLA-C* (major histocompatibility complex, class I, C), *HLA-DRA* (major histocompatibility complex, class II, DR alpha), *HLA-DRB1* (major histocompatibility complex, class II, DR beta 1), *HLA-F* (major histocompatibility complex, class I, F), *HTR2C* (5-hydroxytryptamine receptor 2C), *HTR7* (5-hydroxytryptamine receptor 7), *ICAM1* (intercellular adhesion molecule 1), *ICOSLG* (inducible T cell costimulator ligand), *IFNLR1* (interferon lambda receptor 1), *IL11* (interleukin 11), *IL13RA1* (interleukin 13 receptor subunit alpha 1), *IL15* (interleukin 15), *IL15RA* (interleukin 15 receptor subunit alpha), *IL17RE* (interleukin 17 receptor E), *IL1RL2* (interleukin 1 receptor like 2), *IL24* (interleukin 24), *IL7R* (interleukin 7 receptor), *INHBB* (inhibin subunit beta B), *ITGA10* (integrin subunit alpha 10), *ITGA2* (integrin subunit alpha 2), *ITGA6* (integrin subunit alpha 6), *ITGB2* (integrin subunit beta 2), *ITGB4* (integrin subunit beta 4), *JAG2* (jagged canonical Notch ligand 2), *JUP* (junction plakoglobin), *KCNMB3* (potassium calcium-activated channel subfamily M regulatory beta subunit 3), *KCNMB4* (potassium calcium-activated channel subfamily M regulatory beta subunit 4), *KCNN4* (potassium calcium-activated channel subfamily N member 4), *KISS1* (KiSS-1 metastasis suppressor), *L1CAM* (L1 cell adhesion molecule), *LAMA1* (laminin subunit alpha 1), *LAMA5* (laminin subunit alpha 5), *LAMB1* (laminin subunit beta 1), LPAR2 (lysophosphatidic acid receptor 2), *LRP5* (LDL receptor related protein 5), *MAPK13* (mitogen-activated protein kinase 13), *MGST3* (microsomal glutathione S-transferase 3), *MITF* (melanocyte inducing transcription factor), *MMP1* (matrix metallopeptidase 1), *NCOA1* (nuclear receptor coactivator 1), *NCOA3* (nuclear receptor coactivator 3), *NEO1* (neogenin 1), *NFATC1* (nuclear factor of activated T cells 1), *NGFR* (nerve growth factor receptor), *NRCAM* (neuronal cell adhesion molecule), *P2RY10* (P2Y receptor family member 10), *P2RY2* (purinergic receptor P2Y2), *PCLO* (piccolo presynaptic cytomatrix protein), *PDGFRB* (platelet derived growth factor receptor beta), *PGF* (placental growth factor), *NCAM2* (paired like homeobox 2A), *CFD* (phosphoinositide kinase, FYVE-type zinc finger containing), *PLAU* (plasminogen activator, urokinase), *PLCB1* (phospholipase C beta 1), *PLCD1* (phospholipase C delta 1), *PLCD3* (phospholipase C delta 3), *PLCG1* (phospholipase C gamma 1), *PLCG2* (phospholipase C gamma 2), *PLD2* (phospholipase D2), *PRKCE* (protein kinase C epsilon), *PRKCZ* (protein kinase C zeta), *PRLR* (prolactin receptor), *PROS1* (protein S), PTGER2 (prostaglandin E receptor 2), *PTGER4* (prostaglandin E receptor 4), *PTGS2* (prostaglandin-endoperoxide synthase 2), *RALB* (RAS like proto-oncogene B), *RIMS2* (regulating synaptic membrane exocytosis 2), *RYR2* (ryanodine receptor 2), *S1PR3* (sphingosine-1-phosphate receptor 3), *S1PR5* (sphingosine-1-phosphate receptor 5), *TAP1* (SEC14 like lipid binding 2), *TAP2* (SEC14 like lipid binding 3), *SERPINA1* (serpin family A member 1), *SERPINB2* (serpin family B member 2), *SNAP25* (synaptosome associated protein 25), *SSTR2* (somatostatin receptor 2), *STAT1* (signal transducer and activator of transcription 1), *SUFU* (SUFU negative regulator of hedgehog signaling), *TERT* (telomerase reverse transcriptase), *TGFBR2* (transforming growth factor beta receptor 2), *TNFRSF10D* (TNF receptor superfamily member 10d), *TNFRSF11B* (TNF receptor superfamily member 11b), *TNFRSF19* (TNF receptor superfamily member 19), *TNFRSF1B* (TNF receptor superfamily member 1B), *TNFSF12* (TNF superfamily member 12), *TNC* (troponin C1, slow skeletal and cardiac type), *TRAF3* (TNF receptor associated factor 3), *TRAF5* (TNF receptor associated factor 5), *VCAN* (versican), *VIPR1* (vasoactive intestinal peptide receptor 1), *VSIR* (V-set immunoregulatory receptor), *WNT7B* (Wnt family member 7B).

**Table 3 ijms-26-09611-t003:** Validated primer sequences used for RT-qPCR analysis. Primer pairs targeting genes associated with cell adhesion, mitochondrial function, epithelial–mesenchymal transition (EMT), and housekeeping controls were used to validate transcript expression levels. Sequences were designed based on human gene targets and validated for amplification specificity and efficiency. Gene IDs correspond to NCBI Gene database entries.

Gene ID	Adhesion Genes	Forward Primer Sequence	Reverse Primer Sequence
3688	*ITGB1*	GCCGCGCGGAAAAGATGAAT	CACAATTTGGCCCTGCTTGTA
3673	*ITGA2*	TTAGCGCTCAGTCAAGGCAT	TGCACTGCATAGCCAAACTG
3655	*ITGA6*	GCAGCCTTCAACTTGGACAC	ACGAGCAACAGCCGCTT
8515	*ITGA10*	GGACAGAAACCGATCAGGCA	CCAGGTTAAAGGGGGAGCAG
2335	*FN1*	CGGGACTCAATCCAAATGCC	TTCCAGGAACCCTGAACTGT
3912	*LAMB1*	TGGTTACTGTTGCACACAACG	TTTTGCTTCAGAGACCATCTTGG
1284	*COL4A2*	TGCTTCTGGAAGGGCCAATG	GTCAGTCCCACTTAGCCTCG
1292	*COL6A2*	CCTGCCAAACAGAGCTGTCC	GTGGAAGTTCTGCTCACCCA
3371	*TNC*	TCTCGCCCATCGGAAAGAAAA	GGCTCTAGGGCTCTAGGGTAT
5818	*NECTIN1*	AGCATCCTGCTGGTGTTGAT	TACACGTGCTTCTTGGTGCT
Gene ID	Mitochondrial Genes	Forward Primer Sequence	Reverse Primer Sequence
4512	*MTCO1*	CTTTTCACCGTAGGTGGCCT	AGTGGAAGTGGGCTACAACG
107075141	*MTCO1P12*	CGCCGACCGTTGACTATTCT	TGCCTAGGACTCCAGCTCAT
4535	*MTND1*	TACAACTACGCAAGGCCCC	TGGTAGATGTGGCGGGTTTT
51537	*MTFP1*	AAGAAGGCTGGAGAGGTGCC	TTGATGGTGAAGCCCGGAAT
4519	*MTCYB*	CGCCTTTTCACTAATCGCCC	CGTAATATAGGCCTCGCCCG
4541	*MTND6*	GATCCTCCCGAATCAACCCT	GGGTTAGCGATGGAGGTAGG
Gene ID	EMT Genes	Forward Primer Sequence	Reverse Primer Sequence
59	*ACTA2*	GAGGGAAGGTCCTAACAGCC	TAGTCCCGGGGATAGGCAAA
208	*AKT2*	TGCAGAGATTGTCTCGGCTC	CCGTCACTGATGCCCTCTTT
960	*CD44*	GCACAGACAGAATCCCTGCT	TCTTGCCTCTTGGTTGCTGT
1000	*CDH2*	TGGATGAAAGACCCATCCACG	CAGGGAGTCATATGGTGGAGC
3880	*CK19*	GGGCAACGAGAAGCTAACCA	GGTACCAGTCGCGGATCTTC
1499	*CTNNB1*	AATCAGCTGGCCTGGTTTGA	GCTTGGTTAGTGTGTCAGGC
6615	*SNAI1*	GTTTACCTTCCAGCAGCCCT	TCCCAGATGAGCATTGGCAG
6591	*SNAI2*	GGACCACAGTGGCTCAGAAA	CTTCAATGGCATGGGGGTCT
7431	*VIM*	TCCGCACATTCGAGCAAAGA	AACTTACAGCTGGGCCATCG
Gene ID	Housekeeping Genes	Forward Primer Sequence	Reverse Primer Sequence
3329	*HSPD1*	CGCCGCCGACGACCT	GTGGGTAACCGAAGCATTTCTGC
10376	*TUBA1B*	AGGCCCGTGAAGATATGGCT	AGCTGAAATTCTGGGAGCATGA
506	*ATP5F1B*	TGTCGATCTGCTAGCTCCCT	ACAGAGTAACCACCATGGGC

## Data Availability

The raw data supporting the conclusions of this article will be made available by the authors on request.

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
