# Peer review of "Molecular Adaptations to Repeated Radiation Exposure in Triple-Negative Breast Cancer: Dysregulation of Cell Adhesion, Mitochondrial Function, and Epithelial–Mesenchymal Transition"

_ijms, 2025, doi:10.3390/ijms26199611_

Round 1
Reviewer 1 Report
Comments and Suggestions for Authors The presented study investigates several molecular features of radiation adaptation in TNBC to overcome treatment resistance for irradiation treatments, and presents a well-structured in vitro study that integrates transcriptomics and functional validation to reveal mitochondrial dysfunction, adhesion loss, and EMT as key features of radiation-adapted TNBC cells.
1. [Introduction] "ECM components such as fibronectin, collagen, and laminin interact with integrin receptors expressed on tumor cell surfaces, initiating intracellular signaling cascades that influence tumor cell survival, proliferation, migration, and therapeutic resistance" - The authors should extend the the role of the extracellular matrix beyond ligand–receptor interactions to include mechanotransduction signaling. The mechanotransduction signaling may be influenced by both the biomechanical and biochemical properties of the tumor ECM [https://www.mdpi.com/2673-5261/5/4/29 ], which directly affect both radioresistance and chemoresistance of the tumor. 2. [Methods, Statistical analysis] The paper does not provide an adequate description of the statistical analysis. The sentence presented is clearly insufficient for a bioinformatics paper: "Statistical significance was determined using one-way ANOVA followed by Tukey’s post hoc test in Jamovi. " 3. [Results] The results of the study are based only on the MDA-MB-231 cell line. In the light the TNBC is highly heterogeneous, so findings may not generalize across different TNBC only for conservative single-nucleotide variants/insertions. In addition, adaptation through clonal selection might enrich rare pre-existing resistant clones rather than during routine patient treatment. 4. [Results] The study describes correlations (e.g., downregulated OXPHOS genes, reduced membrane potential), and should restore these features (via knockdown/overexpression or metabolic supplementation) to reduce sensitivity to radiation, for example based on the available current datasets (?). I recommend to differentiate the cell death into the known subtypes - ferroptosis, pyroptosis etc. [https://pmc.ncbi.nlm.nih.gov/articles/PMC10623117/ ], with specific treatment strategies for each cell death type. 5. [Results] Targeting the epithelial-mesenchymal transition through irradiation appears to be ineffective, except for its role in inducing senescent drift in tumor tissue [https://pmc.ncbi.nlm.nih.gov/articles/PMC9989623/ ], especially related mediators of cellular senescence. 6. [Discussion] The manuscript lacks the details on the ways to enhance effectiveness of the up-to-date mRNA therapies for the TNBC, and potential targets of personalized mRNA vaccines beyond the p53 gene [https://pmc.ncbi.nlm.nih.gov/articles/PMC11913391/ ]. However, identifying targets for mRNA therapy was not the aim of the study; nonetheless, the study’s findings could serve as potential targets for mRNA therapies, especially for combined irradiation with chemotherapy.
Reviewer 2 Report
Comments and Suggestions for Authors
In this work, the authors identify a gene profile associated with resistance to repeated radiation courses in MDA-MB-231 triple-negative breast cancer cells. A series of cellular processes are linked to the development of a resistance phenotype. Specifically, resistant cells show a modulation in the expression of adhesion genes. This reduction is connected to changes in EMT markers. The cellular adaptive response has also been associated with altered mitochondrial gene expression and a decreased membrane potential. Overall, I find the authors' results valuable. However, some modifications are required. The authors suggest there may be a connection between adhesion proteins and EMT, but this has not been functionally demonstrated in their work. Additionally, recent research (Cell Biosci. 2024 Dec 30;14(1):156. doi: 10.1186/s13578-024-01336-z) indicates a functional role for the NDRG1 protein in cellular stress response and cell adhesion. This gene is among those differentially expressed. Concerning mitochondrial analysis, it would be beneficial to investigate potential variations in mitochondrial number and shifts in metabolism towards glycolysis or beta-oxidation.
Round 2
Reviewer 1 Report
Comments and Suggestions for Authors
The authors have thoroughly addressed my comments and performed the necessary revisions to the manuscript. I am confident that the revised version represents a valuable and citable contribution to the field of cancer research.
Reviewer 2 Report
Comments and Suggestions for Authors
The manuscript is suitable for publication